# Improving the Technology of Primary Purification of the Safflower Oil Using Secondary Products of Processing on a Biological Basis

**DOI:** 10.3390/foods12173275

**Published:** 2023-08-31

**Authors:** Bauyrzhan Iskakov, Mukhtarbek Kakimov, Rafał Kudelski, Maigul Mursalykova, Amirzhan Kassenov, Zhuldyz Satayeva, Serik Kardenov, Zhanar Kalibekkyzy, Ayaulym Mustafayeva, Aidyn Igenbayev, Michał Bembenek

**Affiliations:** 1The Department of Food Technology and Processing Products, S. Seifullin Kazakh Agrotechnical Research University, Zhenis Avenue 62, Astana 010011, Kazakhstan; 2Faculty of Mechanical Engineering and Robotics, AGH University of Science and Technology, A. Mickiewicza 30, 30-059 Krakow, Poland; 3The Department of Technological Equipment and Machine Engineering, NJSC Shakarim University of Semey, St. Glinka 20A, Semey 071412, Kazakhstan; 4The Department of Food Production Technology and Biotechnology, NJSC Shakarim University of Semey, St. Glinka 20A, Semey 071412, Kazakhstan

**Keywords:** safflower oil, refining, antioxidant, purification, enrichment, filter material, sediment, centrifuge

## Abstract

Safflower oil is a very valuable product for the body and human health. It is rich in macro- and microelements, vitamins and minerals, and also has antioxidant properties. The primary purification of safflower oil is an important stage of its production and directly affects the quality of the final product and its storage ability. Purifying safflower oil using a combination of filtration and sedimentation processes in an experimental cone-shaped centrifuge is a new direction in its processing. The purpose of this study was to determine the effects of flax fiber as a filter material for safflower oil. The Akmai variety of the safflower was tested. The results showed that the quality indicators of safflower oil before and after filtration through flax fiber are different. The amount of unsaturated fatty acids such as oleic (18.31 ± 0.874%) and cis-linoleic acid (82.52 ± 1.854%) increased, as well as the content of arginine (2.1), tyrosine (0.57), methionine (0.4), cystine (2.5), tryptophan (2.6), and other amino acids (in oil g per 100 g of protein). The increase in the total amount of phenols (322.12 ± 6 mgEAG/kg of oil) was observed, which directly caused the higher antioxidant activity (42.65 ± 8%) of the safflower oil. These results demonstrate that flax fiber can enrich safflower oil. To find the optimal conditions for safflower oil centrifugation in a cone-shaped sedimentary-filtering centrifuge, the thickness of the flax fiber and the distance between the inner and outer perforated filter rotor were tested. It was found that the optimal and effective thickness of the flax fiber is 1.5 × 10^7^ nm, while the thickness of the sediment is 0.5 × 10^7^ nm.

## 1. Introduction

Vegetable oils are crucial for human health, being seen not only as sources of energy, but also as essential structural components of the cell membranes. Lipids found in the human body play key roles in various biochemical processes [1,2]. The nutritional value of vegetable oils is significant. Among them, safflower seed oil stands out as a supplier of carbohydrates, proteins, and fats necessary for replenishing energy levels. Additionally, safflower seed oil contains essential fatty acids which are not produced by the human body, which only enhances its significance [3,4].

Safflower (Carthamus tinctorius) is gaining popularity among other oilseeds. Historically, the safflower has been cultivated for its flowers to produce paint, however, its use as an oilseed has grown significantly. Researchers are focused on finding the potential health benefits of safflower oil, including its anticarcinogenic and antiatherosclerotic properties, as well as its ability to activate human body growth without increasing body weight [5,6,7]. The high concentration of linoleic and oleic acids in safflower oil makes it a valuable vegetable oil [8,9,10]. Its ability to normalize cholesterol and glucose levels makes it particularly relevant for peoples suffering from obesity, diabetes, and cardiovascular diseases [11,12]. Moreover, experts recommend including safflower oil in one’s diet to bolster the immune system and to help people suffering from liver, genitourinary, and gallbladder diseases [13,14]. The nutritional value of safflower oil has already been acknowledged in combating and preventing ailments such as hyperlipemia, atherosclerosis, and coronary heart disease [15,16,17]. Moreover, safflower oil contains a considerable energy value, including essential compounds such as tocopherols, phytosterols, phospholipids, and carotenoids [18,19]. However, the oxidative instability of raw safflower oil limits its storage period before processing. The indicator of oxidative stability decreases over time, leading to a reduced shelf life [20,21]. Consequently, safflower oil lacks high resistance to oxidation [22,23,24].

The primary purification of vegetable oils, involving the removal of mechanical impurities, plays a critical role in food production. This purification stage in the manufacturing process of oils significantly impacts the quality indicators. It has also led to the minimization of production and transportation, as well as storage losses [25]. Primary treatment methods such as filtration, settling, and separation can produce a natural product by employing natural filtration materials. Filtration has a direct effect on the subsequent purification processes. During processing, a precipitate is formed in the initial stage and must be subsequently filtered out [26,27,28]. It is essential to recognize that including substances and impurities, rather than triglycerides, determines the color, taste, and smell of oils. Some of these substances, such as phospholipids and vitamins, enhance the nutritional value of fats, while others, such as waxes and gossypol, spoil the quality of fats and cause complications in their technological processing [29,30,31].

Safflower oil, after pressing, becomes a complex, polydisperse liquid containing not only glycerol and related substances, but also protein and non-protein mechanical impurities. Removal of these impurities right after processing is vital to prevent various spoilage processes. Sedimentation, centrifugation, and filtration are commonly used methods for this purpose [31,32].

However, current equipment lacks the capability to simultaneously filter and settle safflower oil. So, it is necessary to develop equipment that can combine both processes through centrifugation [33].

Various filtration methods and materials, including centrifuges, pressure sheets, bags, polishing, cartridge, frame, membrane, and vacuum filters, are used to filter vegetable oils, including safflower oil [34,35,36]. Every purification material and equipment has its advantages and disadvantages when applied to different stages of purification, such as primary purification after pressing or purification after hydration and neutralization. However, all filtering materials are primarily designed to remove mechanical impurities that affect taste, presentation, and promote the oxidation process [37,38].

Despite the potential benefits, the by-products of processing grains and oilseeds are currently not being used as filtering materials. These by-products can be used not only for purifying vegetable oils, but also for enriching them with essential macro- and microelements, vitamins, and other health-benefiting compounds. The goal of this study is to enhance the technology of safflower oil filtration using a cone-shaped sedimentary filter centrifuge. The materials derived from grain processing and the oil and fat industries were used for the deep purification and enrichment of safflower oil. Additionally, the potential to enrich vegetable oils using these filter materials was tested. The Akmai variety of the safflower was tested because it is the most popular variety in Kazakhstan due to its high yield, and its resistance to diseases and climatic conditions. It is grown in Aktobe, Almaty, Zhambyl, Kyzylorda, South Kazakhstan, in the Akmola regions of Kazakhstan. Flax fibers were used as the filtering material. The results can be used for primary purification for not only safflower oil but also other vegetable oils.

## 2. Materials and Methods

### 2.1. Materials

#### 2.1.1. The Material for Extraction of Safflower Oil

Safflower seeds of the Akmai variety were used, which were grown in the Akmola region of Kazakhstan in the experimental sown areas of the S. Seifullin Kazakh Agrotechnical Research University in Astana (Latitude and longitude coordinates are: 51.169392, 71.449074). The fat content in absolutely dry seeds is 36–37%, and in the kernel it is 55.0–55.7% [39].

#### 2.1.2. Filtering Material

As a filtering material, secondary processing products of the grain processing and oil and fat industries, such as wheat bran (Figure 1a), rye bran (Figure 1b), oat bran (Figure 1c), and flax fiber (Figure 1d), were preliminary choose. The information about materials (Table 1) was obtained from the bran producers of the LLP “AL and KS” and information about flax fibers was obtained from the developers of the NJSC National Agrarian Scientific and Educational Center [40]. 

All the secondary products are rich sources of proteins, fats, carbohydrates, and dietary fiber, as well as essential vitamins B and E, and important minerals such as iron, zinc, and others. These components offer various health benefits for the human body. For instance, wheat bran is known to be beneficial for the gastrointestinal tract and can help reduce the risk of developing cardiovascular diseases (CVD), cancer, and metabolic disorders. Rye bran, on the other hand, is abundant in dietary fiber, which promotes the growth of beneficial bacteria in the large intestine, thus preventing the development of dysbacteriosis. Furthermore, oat bran contains a significant amount of fiber in the form of β-glucan, which has been shown to reduce cholesterol levels by up to one-third and plays a preventive role against atherosclerosis [41,42,43,44,45,46,47,48].

The preliminary filtration tests showed that the best filtering material turned out to be flax fiber. Further testing was performed on this filtering material only.

### 2.2. Methods

#### 2.2.1. The Filtration Tests in the Experimental Cone-Shaped Sediment Filter Centrifuge

The cone-shaped sediment filter centrifuge used for this test was developed by Iskakov Bauyrzhan, Kakimov Mukhtarbek, Tokhtarov Zhaiyk, Mursalykova Maygul, Sataeva Zhuldyz. It was designed for the deep purification of mechanical impurities from vegetable oils. The equipment design is registered and protected with the patent for invention No. 36262 of the Republic of Kazakhstan (issued from 15 September 2023). The experimental cone-shaped sediment filter centrifuge was based on the Molmash YA7-OCM model (Molmash, Barnaul, Russian Fereration), (Figure 2). The experimental cone-shaped sediment-filtering centrifuge is equipped with a funnel (1) for loading the safflower oil, connected to the main body of the centrifuge through the loading fitting (2) and fixed with a bolt (3) [49]. The funnel (1) for loading the safflower oil is connected to the main body of the centrifuge through the loading fitting (2), which is separated from the flange (9) of the main body. The main working part is located in the separator, which contains an external perforated rotors for filtration (10) and an internal perforated rotor for settling (11). The safflower oil and filter material (15) are placed between them for the additional filtration and enrichment of the safflower oil. The perforations are in the shape of round holes on the surface of the rotor. The holes in the rotor are different; the outer diameter is 0.3 mm and the inner diameter is 0.5 mm. The holes of the inner rotor are larger to avoid plugging the hole and also to make cleaning the safflower oil through the filter media easier. In the filtration material, all impurities are deposited and purified, and the enriched safflower oil exits through the outer rotor. The rotors are placed by a drive coupling and a seal (14) on the drive motor shaft (7), which is protected by a casing (8) to minimize noise in the workplace. In the process of the sedimentation and filtration of the safflower oil, the dehydrated fraction is where the sediment passes through the tray (12) to the outside, while the purified safflower oil exits through the lower fitting (13). The centrifuge is closed with a cover (4) and secured with locks (5).

The nominal productivity of the used centrifuge is 900 kg/h, the capacity of the working chamber is 9 L and is made of stainless-steel grade 12X18H10T, the size of the filter mesh openings is 0.5 mm, and the rotation speed of the rotor shaft is 2850 RPM. Figure 3 shows the process of refining the safflower oil, as well as the principle of operation for the main part of the sediment filter rotor.

From Figure 3 it can be seen that the safflower oil enters the working chamber and is thrown onto the inner filter rotor due to centrifugal force, and the settling process takes place. The safflower oil passes through the filtering material and the outer rotor. In the end, the sediment exits through the top of the rotor and is discharged through the tray, and the filtrate (the safflower oil) exits through the fitting. 

Filtering material 10, 15, 20, and 25 mm thick was placed between the outer and inner rotors and tested. Once 1 L of the oil was dosing to the centrifuge and filtering, sediment thickness, filter material thickness, rotor shaft speed on output, and safflower oil separation factor were determined. 

#### 2.2.2. Density and Specific Gravity of the Safflower Oil

Density and specific gravity were determined using the AON-1 hydrometer of the JSC “Steklopribor” company (Zavodskoe, Ukraine). The refractive index of the safflower oil was determined with the Atago PAL-RI series refractometer (Tokyo, Japan). Method of determination was carried out according to GOST 18848-2019 [50].

#### 2.2.3. Determination of the Viscosity of the Safflower Oil

To measure the viscosity of the safflower oil, a Brookfield model DV-E viscometer (Milwaukee, WI, USA) was used and the analyses were carried out according to GOST 33768-2015 [51].

#### 2.2.4. The Chemical Properties of the Safflower Oil

The chemical properties were determined according to AOAC (2005) [52] and are iodine value, acid value, peroxide value, and saponification value of safflower oil.

#### 2.2.5. The Composition of Fatty Acids

The fatty acid composition of safflower oil was analyzed in accordance with GOST 30623-2018 [53], which specifies the method for determining the fatty acid composition of vegetable oils and blended fat products using an Agilent 7890A chromatograph (Agilent Technologies, Penang, Malaysia). In addition to determining the fatty acid composition, the amount of α-, β-, γ-, and δ-tocopherols in safflower oil was measured using a high-performance liquid chromatography (HPLC) method. The inorganic composition of the pressed safflower oil was assessed following the relevant state standards of the Republic of Kazakhstan. This analysis involves determining the levels of various essential minerals and elements in the oil, such as iron, zinc, and others. 

The combination of gas chromatography and HPLC in this study ensures a comprehensive analysis of safflower oil, encompassing its fatty acid profile, tocopherol content, and inorganic composition. Pressed safflower oil was chosen as a starting point, since it is more natural and richest in macro- and microelements, vitamins, and other beneficial substances for the human body. In this case, small and large mechanical impurities are not counted in the composition of the oil that is removed in the process of primary oil purification through the filtration and settling.

#### 2.2.6. The Composition of Inorganic Substances

The composition of inorganic substances was investigated using the following regulatory documents:

GOST 30538-97 [54], GOST R 51766-2001 [55], GOST R 53183-2008 [56], GOST R 52676-2006 [57], GOST 31905-2012 [58], GOST R 55484-2013 [59].

#### 2.2.7. The Mass Fraction of Non-Fat Impurities and Sediment

The mass fraction of non-fat impurities and sediment was determined using GOST 5481-2014 [60]. 

#### 2.2.8. Determining the Amount of Amino Acids

The amino acid composition was determined using the Kapel-105 (LUMEX Company, Saint Petersburg, Russia) capillary electrophoresis system (MVI M 04-38-2004).

#### 2.2.9. The Total Amount of Phenols

The total amount of phenols in the safflower oil was analyzed according to the method described by Merouane et al. [61].

#### 2.2.10. Determination of Antioxidant Activity

Determination of the antioxidant activity of the safflower oil was carried out according to the method of Nogala-Kalucka et al. [62].

#### 2.2.11. Measuring the Filtration Material Thickness

The spiral micrometer thickness Gauge, 0–12.7 mm range, (Mitutoyo Corporation, Tokyo, Japan) Randomly selected five points on the material that were measured. 

## 3. Results and Discussion

### 3.1. Physical Indicators of Safflower Oil

The physical indicators of the safflower oil are shown in Table 2. Each value is the average of three determinations.

Analyzing the results from Table 2, it can be seen that the density of the safflower oil before filtration was 0.922 ± 0.196 g/mL, and after filtration, a slight decrease in density to 0.884 ± 0.192 g/mL can be seen. The specific gravity of the safflower oil before filtration was 0.931 ± 0.197, and due to oil purification and an increase in the proportion of polyunsaturated fatty acids, the specific gravity of the safflower oil increased to 0.965 ± 0.201. For quantitative identification, the determination of the specific gravity and refractive index of the oil does not provide enough information, however, these data are useful for determining adulteration and contamination of the oil [63]. From the research, it should be noted that the more linoleic and linolenic acids belonging to the group of polyunsaturated fatty acids, the higher the refractive index. The refractive index also reflects the quality, purity of the oil, and the degree of its oxidation during storage. The refractive index of safflower oil before filtration was 1.4771 ± 0.248 and after filtration it increased to 1.4812 ± 0.485. Slight changes were observed in the viscosity of the safflower oil: before filtration it was 46.6 ± 1.393 cP and after filtration it was 46.8 ± 1.397 cP.

### 3.2. Chemical Indicators of the Safflower Oil

The iodine value, acid value, saponification value, and peroxide value of the safflower oil were determined, and the results are shown in Table 3. Each value is the average of three determinations.

Table 3 shows that the acid number of safflower oil before filtration was 1.066 ± 0.211 mg KOH/g and after filtration the acid number increased slightly to 1.075 ± 0.212 mg KOH/g. The iodine number of safflower oil before filtration was 148.19 ± 2.485 g in I2/100 g, while after filtration it was 150.15 ± 2.501 g in I2/100 g, which confirms the increase in the amount of unsaturated fatty acids. The peroxide number before filtration was 8.09 ± 0.581 mol/kg and after filtration it became 7.01 ± 0.540 mol/kg, which is a good indicator of storage capacity. The saponification number also indicates the amount of potassium hydroxide; however, the saponification number determines the amount of ester bonds in the fat. The saponification number of safflower oil before filtration was 160.7 ± 2.588 mg KOH/g and after filtration it was 161.5 ± 2.594 mg KOH/g.

### 3.3. Fatty Acid Profile, Total Inorganic Matter and Tocopherols in Safflower Oil

The fatty acid composition of pressed safflower oil was studied on chromatography by gas chromatography of mass fraction methyl esters of fatty acid (Table 4).

The analysis of the fatty acid composition of safflower oil shows information about the saturated fatty acids in safflower oil: C16:0 palmitic acid prevails (7.6 ± 0.563%), followed by C18:0 stearic acid (2.56 ± 0.327%), C8:0 caprylic acid (0.94 ± 0.198%), and C14:0 myristic acid (0.35 ± 0.121%). However, after filtration of the safflower oil, the decrease in the amount of saturated fatty acids by almost two times can be seen, and there is no C8:0 caprylic acid. In the case of each fatty acid, the content of C16:0 palmitic acid was 4.34 ± 0.425%, C18:0 stearic acid was 1.81 ± 0.275%, and C14:0 myristic acid was 0.07 ± 0.054%. Table 4 shows a clear quantitative overweight of unsaturated fatty acids over saturated fatty acids, and what is most important, after filtration, the concentration of unsaturated fatty acids increased significantly. However, there are exceptions where the content of individual fatty acids decreased. In the case of a detailed description of each group of unsaturated fatty acids it will be necessary to consider mono- and polyunsaturated fatty acids separately. For the monounsaturated fatty acids, the content of C16:1 palmitoleic acid (0.05 ± 0.046%) and C20:0 arachidic acid (0.30 ± 0.112%) decreased. On the other hand, the second group of monounsaturated fatty acids such as C18:1 oleic acid (18.31 ± 0.874%), C22:0 behenic acid (0.24 ± 0.100%), C17:1 cis-10-heptadecenoic acid (0.23 ± 0.098%), and C20:1 gadoleic acid (0.11 ± 0.068%) doubled in content. The polyunsaturated fatty acids predominate over others before and after filtration; for example, the content of C18:2 cis-linoleic acid before filtration was 74.82 ± 1.766%, while after it became 82.52 ± 1.854%. However, the concentration of C18:3 a-linolenic acid decreased from 0.95 ± 0.199% to 0.22 ± 0.096%. These results are quite similar to those of Katkade et al. [64]. 

The inorganic composition of safflower oil before and after filtration changed. The concentration of many elements increased greatly, even though the amount of sulfur was 0.085 ± 0.059 mg/kg, after filtration it was not found in the composition of the safflower oil. The rest of the content of the inorganic substances is as follows: iron (0.024 ± 0.032 mg/kg) 0.045 ± 0.043 mg/kg, phosphorus (0.067 ± 0.053 mg/kg) 0.346 ± 0.120 mg/kg, silicon (0.066 ± 0.052 mg/kg) 0.249 ± 0.102 mg/kg, chlorine (0.091 ± 0.062 mg/kg) 0.466 ± 0.139 mg/kg, and calcium (0.107 ± 0.067 mg/kg) 0.236 ± 0.099 mg/kg increased at least twice.

Additionally, from Table 4, it can be seen that the total content of α-tocopherols in the safflower oil rises from 508.26 ± 4.602 mg/kg to 556.7 ± 4.816 mg/kg after filtration. This indicates the high resistance of safflower oil to oxidation, despite the significant amount of linolenic acid. The amount of tocopherols in safflower oil is consistent with the results of a study by Elif Aksoz et al. [65].

### 3.4. The Mass Fraction of Non-Fat Impurities and Sludge

Non-fat impurities and sludge are contaminants that must be removed during processing to prevent adverse reactions or undesirable changes in the appearance of the oil from occurring. The results are shown in Table 5.

The mass fraction of non-fat impurities and sludge according to the results before filtration was 0.45 ± 0.13% and after filtration it was 0.02 ± 0.02%, which is within the normal range of high-quality filtration pressed safflower oil.

### 3.5. Amino Acids in Safflower Oil

The amino acid composition of safflower oil before and after filtration is shown in Figure 4. In general, safflower oil is not rich in amino acids, but they are still present in the composition. After filtration, a significant increase in the level of amino acids in safflower oil can be observed, e.g., the level of arginine 1.16, tyrosine 0.17, methionine 0.17, glutalic acid 0.15, and aspartic acid 0.11 increased two or more times, respectively, by 2.1, 0.57, 0.4, 0.32, and 0.41. Other amino acids also increased, and this shows the positive effect of the flax fiber not only as a filtering material, but also as a source of enrichment. Analyzing the results of the work of Funda Arslanoğlu and Selim Aytaç, flax is rich in amino acids, from which it can be assumed that some components were transferred during the filtration process [66]. In addition, it is necessary to pay attention to the high content of cystine 2.5 and tryptophan 2.6; after filtration, arginine 2.1 caught up to their level.

### 3.6. Total Phenols and Antioxidant Activity of the Safflower Oil

To determine the antioxidant activity of the safflower oil, an analysis of the total amount of phenols was carried out (Table 6). Since the amount of phenols, unsaturated fatty acids, and vitamins directly affects the antioxidant activity of safflower oil in the human body, the higher the indicators, the higher the antioxidant activity [67].

The antioxidant activity of the safflower oil extract before filtration was around 41% and after filtration a slight increase up to 42% can be observed (Table 6). This is a fairly high indicator for the antioxidant activity of the safflower oil, since, for comparison, in the works of Zemour et al. [36], the highest total phenols and antioxidant activity were 412.8 ± 1.3 and 68.9 ± 0.4b in Syrian safflower oil of the 2017 harvest, and the lowest was 168.1 ± 7.1b and 24.7 ± 0.7b for the 2015 harvest of Algerian safflower oil. In this article the authors also describe the direct effect of the total amount of phenols on the antioxidant activity of safflower oil, as well as a direct positive effect on human health.

### 3.7. Analysis of Data Dependences of the Technological Process of Centrifugation of the Safflower Oil on a Cone-Shaped Sedimentary-Filtering Centrifuge

To simplify the mathematical description of the purification process in a cone-shaped sediment-filtering centrifuge, it can be considered as a two-stage process:(1)the centrifugation process, which is the purification of the liquid when it is displaced by particles of impurities along the radius to the inner shell of the cone;(2)the process of liquid (oil) movement in the direction along the generatrix of the cone, while impurities are deposited in the pores of the filtering material, as well as on the walls of the inner part of the rotor.

The nature of the flow of centrifugal filtration is determined by the centrifugation mode, i.e., the centrifuge speed given by the thickness of the filter layer in the rotor, the method of loading the rotor with a suspension, the ratio between the solid and liquid phases in the material being processed, and the physicochemical properties of the latter. In the theoretical and practical research, the relationship between oil yield, separation factor, filter media thickness, and sediment thickness can be found. The last three factors directly affect the yield of refined safflower oil. In the process of centrifugation, the separation factor takes a special place and is one of the main indicators. Based on the above studies, flax fiber was used as the filter material. In Figure 5, the sedimentation filter rotor can be seen, where the safflower oil is filtered and settled. On the rotor, the thickness of the filter material can be changed, i.e., flax fiber, as necessary by adjusting the distance between the inner and outer perforated filter rotors. Through this function, the filtration path is being changed, which affects the filtration quality, the safflower oil yield, and the separation factor. In addition, these parameters are directly affected by the thickness of the sediment that is deposited on the internal perforated rotor. By adjusting the distance between the inner and outer perforated rotors, as well as the thickness of the flax fiber, the safflower oil can be obtained from a primary to a highly refined purification grade. At the same time, the quality of the incoming press oil, which can facilitate or complicate the filtration process, cannot be ignored.

When studying the technological process of centrifugation on a cone-shaped centrifuge, the results presented in Figure 6 were obtained.

In Figure 6a,b the effect of flax fiber thickness and sediment on the safflower oil yield and separation factor can be seen. In Figure 6a, the yield of the safflower oil rises from 700 L/h to 800 L/h, after which it begins to fall. While the thickness of the flax fiber decreases, the thickness of the sediment increases linearly, thereby reducing the oil yield. A low oil yield is obtained, 300 L/h, with a flax fiber thickness of 2.5 × 10^7^ nm, and a draft of 1 × 10^7^ nm. On the other hand, a high oil yield is observed with a flax fiber thickness of 1.5 × 10^7^ nm and 0.5 × 10^7^ nm of sediment. In Figure 6b, at the beginning of the cleaning process, the separation factor is low, around 500, after which it begins to grow up to 1500. The highest separation factor is reached at a thickness of 2 × 10^7^ nm, while a low separation factor is observed at a flax fiber thickness of 1 × 10^7^ nm, 0.1 × 10^7^ nm of sediment. In Figure 6c,d, the effects of flax fiber thickness and sediment thickness on oil yield and separation factor are shown. An almost similar result is noticed, as the thickness of the flax fiber and the thickness of the sediment have the same effect on the oil yield and the separation factor. With a flax wire thickness of 2 × 10^7^ nm and a sediment thickness of 7 × 10^6^ nm, in both graphs, the oil yield is 700 L/h, while the safflower oil separation factor is 800. With a low oil yield of 400 L/h and a separation factor of 1600, a flax fiber thickness of 2 × 10^7^ nm and a sediment thickness of 0.7 × 10^7^ nm are noticed.

In conclusion, the thickness of the flax fiber and sludge directly affect the oil yield and separation factor. As a result of this research, the optimal conditions for centrifuging safflower oil to remove mechanical impurities are a flax fiber thickness of 1.5 × 10^7^ nm and a sludge thickness of 0.5 × 10^7^ nm. The oil yield and separation factor are optimal and efficient then.

## 4. Conclusions

Vegetable oils, including safflower oil, are gaining popularity as versatile products for both food and technical applications. The rich concentration of essential unsaturated fatty acids, such as linoleic and oleic acids, contributes significantly to the nutritional value and health-promoting properties of safflower oil. Its properties are known to enhance immunity and aid in the prevention and treatment of various diseases.

Filtration, as a crucial primary purification process, plays a significant role in improving the quality of the safflower oil. It directly impacts the oil’s quality indicators and shelf life, while also streamlining subsequent purification processes. The successful implementation of effective filtration techniques can reduce production costs and lead to an overall improvement in the final product’s quality.

The use of secondary processing products from grain processing and the oil and fat industries as filter materials represents a promising and innovative approach. In this regard, the experimental application of flax fiber as a filter material for safflower oil filtration has shown promising results. Flax fiber has been found to be the most effective and optimal filter material, positively influencing the physical and chemical parameters of safflower oil. The use of flax fiber has led to notable improvements in the fatty acid composition, tocopherol, and phenol content, consequently enhancing the antioxidant activity of safflower oil. These findings serve as a solid scientific basis for further research in the area of vegetable oil enrichment using secondary processing products, as well as their utilization as filter materials.

Beyond the technological aspect, the investigation of the centrifugation process has yielded valuable insights. The thickness of the flax fiber and the sediment have direct implications for essential parameters such as safflower oil yield and separation factor. Adjusting the distance between the inner and outer rotors, where the filtering material is positioned, parameters such as 1107 nm, 1.5107 nm, 2107 nm, and 2.5107 nm were explored. These adjustments directly impacted the separation factor, oil yield, and sediment thickness formed on the walls of the inner rotor. These findings provide a valuable framework for optimizing the centrifugation process of vegetable oils to effectively remove mechanical impurities and obtain naturally enriched food products.

In conclusion, the research presented in this study highlights the potential for improving safflower oil production through enhanced filtration processes using secondary processing products as filter materials. The successful application of flax fiber as a filter material has demonstrated significant improvements in the quality and nutritional profile of safflower oil. These findings hold promising implications for the food industry in producing high-quality and enriched vegetable oils, thus contributing to human health and well-being.

## 5. Patents

Iskakov Bauyrzhan Myrzabekovich, Kakimov Mukhtarbek Mukanovich, Tokhtarov Zhaiyk Khamitovich, Mursalykova Maygul Taurzhanovna and Sataeva Zhuldyz Isakovna. Centrifuge for deep purification of vegetable oils from mechanical impurities. For invention patent No. 36,262, filed 9 March 2022, and issued 15 September 2023.

## Figures and Tables

**Figure 1 foods-12-03275-f001:**
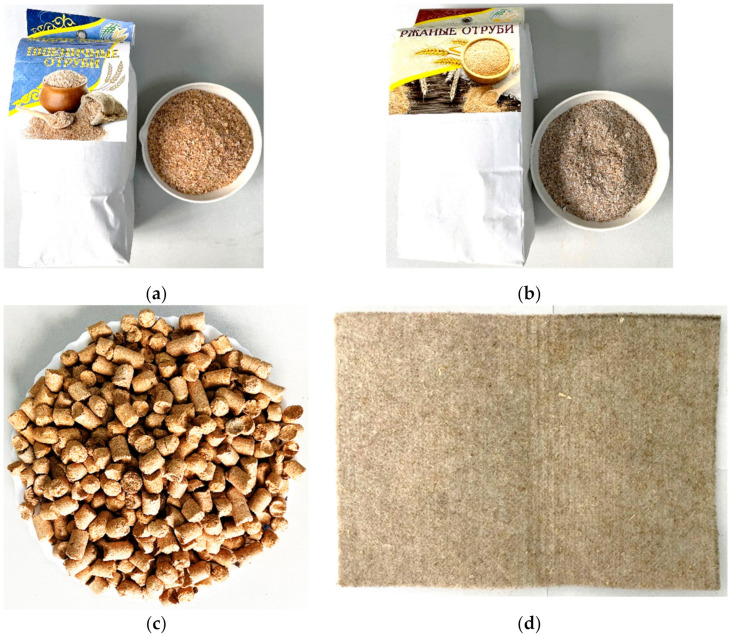
Secondary products of agricultural processing; (**a**) wheat bran; (**b**) rye bran; (**c**) oat bran; (**d**) flax fiber. NJSC “National Agrarian and Scientific Educational Center”.

**Figure 2 foods-12-03275-f002:**
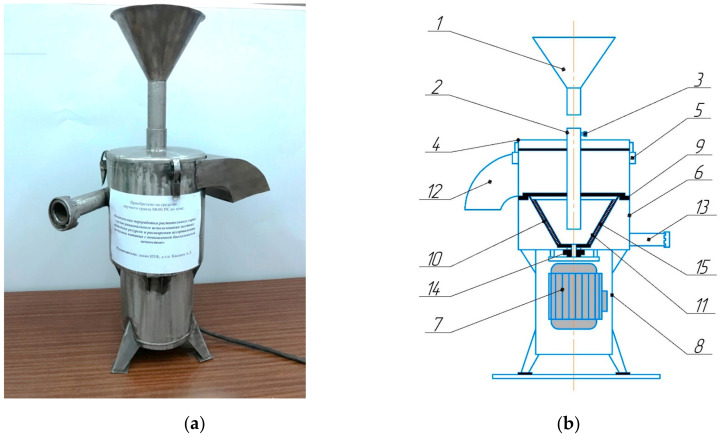
Experimental cone-shaped sediment filter centrifuge; (**a**) picture of a centrifuge; (**b**) centrifuge drawing; 1—load funnel; 2—boot fitting; 3—bolt for fastening the loading funnel; 4—lid centrifuge; 5—locks for fastening the centrifuge cover; 6—separator body; 7—drive motor; 8—the protective casing of the electric motor; 9—separating flange; 10—external perforated filter rotor; 11—internal filtering rotor; 12—tray output of the dehydrated fraction; 13—pipe oil outlet fitting; 14—drive coupling and seal; 15—filter material.

**Figure 3 foods-12-03275-f003:**
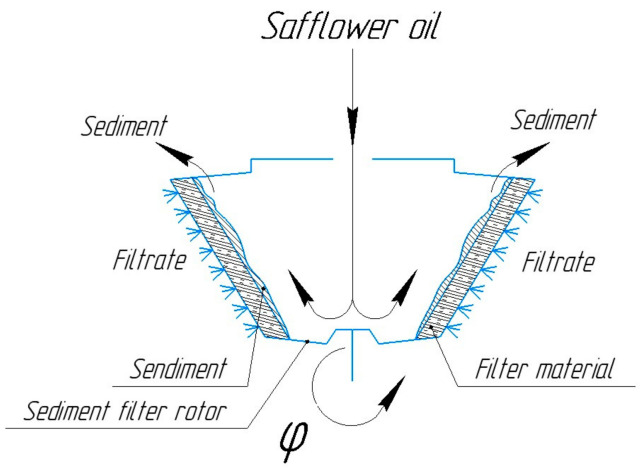
The principles of operation for the sediment filter rotor.

**Figure 4 foods-12-03275-f004:**
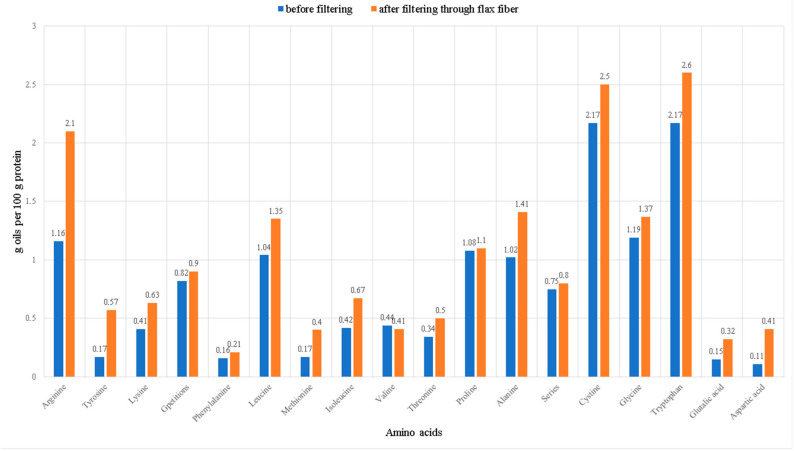
Amino acid composition of safflower before and after filtration (g oils per 100 g protein).

**Figure 5 foods-12-03275-f005:**
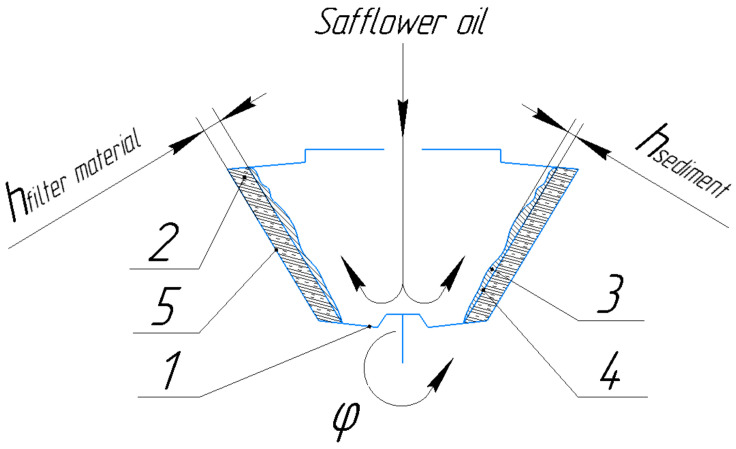
Sediment-filtering rotor: 1—sedimentary-filtering rotor; 2—filter material (flax fiber); 3—precipitate; 4—inner perforated filter rotor; 5—external perforated filter rotor.

**Figure 6 foods-12-03275-f006:**
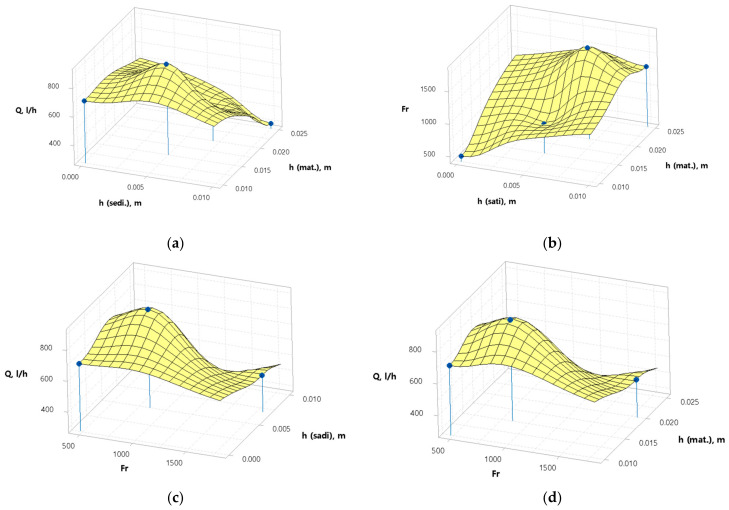
The safflower oil yield graph; (**a**) influence of sediment thickness (h_sadi_) and filter material thickness (F_mat_) on oil yield (Q); (**b**) influence of sediment thickness (h_sadi_) and filter material thickness (h_mat_) on the separation factor (Fr); (**c**) effect of separation factor (Fr) and sediment thickness (h_sadi_) on oil yield (G); (**d**) effect of separation factor (Fr) and filter material thickness (h_mat_) on oil yield (G). **Note:** Graphs were built in meters, while the thickness of the filter material and sediment were measured according to the method.

**Table 1 foods-12-03275-t001:** Chemical composition of the secondary products of agricultural processing.

Secondary Products	Protein, g	Fats, g	Carbohydrates, g	Alimentary Fiber, g	Vitamins			Minerals	
					B1, %	B6, %	E, %	Fe, %	Zn, %
Wheat bran	14.7	4.1	23.5	43.6	50	65	69	78	61
Rye bran	15.5	3.8	17	43.6	36	21	10	56	36
Oat bran	17.3	7	50.8	15.4	78	8.3	6.7	30	26
Flax fibers	16	4.3	64.5	43.6	50	9	69	78	59

**Table 2 foods-12-03275-t002:** Physical characteristics of safflower oil.

	Physical Indicators	
Index	Before Filtration	After Filtering through Flax Fiber
Density (g/mL)	0.922 ± 0.196	0.884 ± 0.192
Specific gravity	0.931 ± 0.197	0.965 ± 0.201
Refractive index	1.4771 ± 0.248	1.4812 ± 0.485
Viscosity (cP)	46.6 ± 1.393	46.8 ± 1.397

**Table 3 foods-12-03275-t003:** Chemical characteristics of safflower oil.

	Chemical Indicators	
Index	Before Filtration	After Filtering through Flax Fiber
Acid number (mg KOH/g)	1.066 ± 0.211	1.075 ± 0.212
Iodine number (g in I2/100 g)	148.19 ± 2.485	150.15 ± 2.501
Peroxide value (mol/kg)	8.09 ± 0.581	7.01 ± 0.540
Saponification number (mg KOH/g)	160.7 ± 2.588	161.5 ± 2.594

**Table 4 foods-12-03275-t004:** The fatty acid composition of the safflower oil.

The Name of the Indicators	Characteristics and Norms, %	Results before Filtering, %	Results after Filtering through Flax Fiber, %
**Saturated fatty acids**			
Caprylic acid C8:0	not standardized	0.94 ± 0.198	-
Myristic acid C14:0	up to 1.0	0.35 ± 0.121	0.07 ± 0.054
Palmitic acid C16:0	2.0–10.0	7.6 ± 0.563	4.34 ± 0.425
Stearic acid C18:0	1.0–10.0	2.56 ± 0.327	1.81 ± 0.275
**Monounsaturated fatty acids**			
Palmitoleic acid C16:1	up to 0.5	0.08 ± 0.058	0.05 ± 0.046
Oleic acid C18:1	7.0–12.2	12.13 ± 0.711	18.31 ± 0.874
Arachic acid C20:0	up to 2.5	0.33 ± 0.117	0.30 ± 0.112
Behenic acid C22:0	up to 0.5	0.1 ± 0.065	0.24 ± 0.100
Cis-10-heptadecenoic acid C17:1	not standardized	0.1 ± 0.065	0.23 ± 0.098
Gadoleic acid C20:1	not standardized	0.06 ± 0.050	0.11 ± 0.068
**Polyunsaturated fatty acids**			
Cis-linoleic acid 18:2	55.0–81.0	74.82 ± 1.766	82.52 ± 1.854
α-linolenic acid18:3	up to 1.0	0.95 ± 0.199	0.22 ± 0.096
**Inorganic composition: (mg/kg, no more)**			
Iron	5.0	0.024 ± 0.032	0.045 ± 0.043
Phosphorus	not standardized	0.067 ± 0.053	0.346 ± 0.120
Silicon	not standardized	0.066 ± 0.052	0.249 ± 0.102
Sulfur	not standardized	0.085 ± 0.059	-
Chlorine	not standardized	0.091 ± 0.062	0.466 ± 0.139
Calcium	not standardized	0.107 ± 0.067	0.236 ± 0.099
Tocopherols (mg/kg)	230–660	508.26 ± 4.602	556.7 ± 4.816

**Table 5 foods-12-03275-t005:** The mass fraction of non-fat impurities and sludge in the safflower oil.

The Name of the Indicators	Characteristics and Norms	Results before Filtering	Results after Filtering through Flax Fiber
Mass fraction of non-fat impurities and sludge: (%, no more)	0.1 ± 0.06	0.45 ± 0.13	0.02 ± 0.02

**Table 6 foods-12-03275-t006:** The total polyphenols and antioxidant activity in safflower oil.

Vegetable Oil	Total Phenols before Filtration (mgEAG/kg of Oil)	Antioxidant Activity (%) before Filtration	Total Phenols after Filtration (mgEAG/kg of Oil)	Antioxidant Activity (%) after Filtration
The safflower oil from Akmay seeds	310.5 ± 8.4	41.12 ± 2.7	322.12 ± 6	42.65 ± 8

In the same column for each year, means with the same letter were not significantly different at *p* < 0.05.

## Data Availability

The data used to support the findings of this study can be made available by the corresponding author upon request.

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
