# Peer review of "Improving the Technology of Primary Purification of the Safflower Oil Using Secondary Products of Processing on a Biological Basis"

_foods, 2023, doi:10.3390/foods12173275_

Round 1

Reviewer 1 Report

Regardless of the originality of the idea, I will propose to reject the paper because the authors did not follow the instructions for the authors and did not scientifically present the results. Starting with the title, which needs to be corrected, the materials and methods, which are not adequately described, the lack of statistical processing of the data, the inadequate and unclear presentation of the results and the inadequate conclusion. With such a good idea and results, rearrange the manuscript, follow the authors' instructions, do the statistics, and get help from someone in English, and try again.

Extensive editing of the English language is required.

Author Response

Dear Reviewer,

Thank you very much for taking the time to carefully read our manuscript. We have accurately read all the comments and referred to all of them. They helped us significantly improve the article.

This is now a completely different paper from that one we originally submitted to the journal. We hope that it will now meet the standards and receive your recommendations for publication. We do not describe the individual small comments contained in the pdf file from you, because in our opinion it does not make sense. Below are general responses to your comments.

Remark 1:

The title needs to be corrected

Answer 1:

Thank You for this remark. On your recommendation, we have changed the topic of the article:

We changed the title of manuscript in accordance with your suggestion. From "Improving the Technology of Refining the Safflower Oil Using the Flax Fiber for the Primary Purification" to "Improving the technology of primary purification of safflower oil using secondary products of processing on a biological basis".

Remark 2:

Materials and methods are not properly described

Answer 2:

Thank you for that comment. We have reviewed the methods and materials, made changes and corrected them.

Remark 3:

Ask for help from someone in English.

Answer 3:

Thank you for this comment. We have revised our article and reworked it.

We have reworked the entire article and made adjustments in order to improve the quality of the English language.

Note: The Reference section has been supplemented and corrected.

Reviewer 2 Report

The authors have submitted a manuscript in which they used a cone-shaped sedimentary filter centrifuge to improve the technology of filtration of safflower oil. Moreover, secondary products from the grain processing and oil and fat industries were used as a filter material for the deep purification and enrichment of safflower oil.

The topic is interesting for the Journal, the results are presented quite well, although I think the article can be further improved to make it clear, especially in the English writing.  

I’d change the title in order to clearly explain what is your main innovation (from a technological point of view). Furthermore, the novelty is not well explained. It takes a careful analysis of the existing literature in this topic to understand the limits of the current technology and in which way you will ameliorate them with your innovation.

Use always the same number of significant digits and choose this number appropriately.

Fig.4 y axis not clear

Results of the antioxidant activity cannot be in %. Use a reference to help understand (e.g., Trolox).

Discussion of results is too generic needs a huge improvement. Moreover, the analysis and the explanation of the different behaviour of the various filtering materials deserve further discussion.

Conclusions should be improved and shortened. They must not be a repetition of the results obtained, except in short. It would be interesting to understand what are the next steps to be taken in view of a larger scale application.

Other minor remarks:

Use always mL instead of ml

Line 396 Do not use “let’s start”, too colloquial

Check the citation reference within the text, accordingly to Journal indication

The English form needs significant improvement as the manuscript is challenging to read

Author Response

Thank you very much for taking the time to carefully read our manuscript and give recommendations for its correction and improvement. We have carefully read the comments and referred to all your comments.

Remark 1:

The article can be improved to make it more understandable, especially in English.

Answer 1:

Thank You for this remark. The English of manuscript has been improved.

Remark 2:

I’d change the title in order to clearly explain what is your main innovation (from a technological point of view).

Answer 2:

Thank You for this remark. We changed the title of manuscript in accordance with your suggestion. From "Improving the Technology of Refining the Safflower Oil Using the Flax Fiber for the Primary Purification" to "Improving the technology of primary purification of safflower oil using secondary products of processing on a biological basis".

Remark 3:

The novelty is not well explained. It takes a careful analysis of the existing literature in this topic.

Answer 3:

Thank You for this remark. We have carefully analyzed the existing literature to give a more complete explanation of the novelty of our research and have rewritten the introduction. In addition, we drew attention to the limitations of modern filtration technologies and talked about how our proposed innovation eliminates these limitations to improve the process of primary purification of safflower oil.

Introduction

Vegetable oils are crucial for human health, serving not only as a source of energy but also as essential structural components of cell membranes. Lipids found in the human body play pivotal roles in various biochemical processes [1,2]. The nutritional value of vegetable oils is unique, and among them, safflower seed oil stands out as a prominent supplier of carbohydrates, proteins, and fats necessary for replenishing energy levels. Additionally, safflower seed oil carries essential fatty acids that are not produced by the body, further enhancing its significance [3,4].

Safflower (Carthamus tinctorius) is gaining popularity among oilseeds. Historically, safflower has been cultivated for its flowers to produce paint, but in recent times, its use as an oilseed has grown significantly. Researchers have become increasingly interested in the potential health benefits of safflower oil, including its anticarcinogenic and anti-atherosclerotic effects and its ability to activate growth with a slight increase in body weight [5-7]. Notably, safflower oil contains a high concentration of linoleic acid, an unsaturated fatty acid known to help lower blood cholesterol levels [8].

The high concentration of linoleic and oleic acid in safflower oil makes it an exceptionally valuable vegetable oil [9,10]. Its potential to normalize cholesterol and glucose levels makes it particularly relevant for individuals with obesity, diabetes, and cardiovascular diseases [11,12]. Moreover, experts recommend incorporating safflower oil into diets to bolster the immune system and to benefit those prone to liver, genitourinary, and gallbladder diseases [13,14].

In addition to its nutritional benefits, safflower oil possesses considerable energy value, containing essential compounds like tocopherols, phytosterols, phospholipids, and carotenoids [15-16]. However, the oxidative instability of raw safflower oil limits its storage time before processing. The induction period, an indicator of oxidative stability, decreases over time, leading to reduced shelf life [17, 18]. Consequently, safflower oil lacks high resistance to oxidation [19-21].

Primary purification of vegetable oils, involving the removal of mechanical impurities, plays a critical role in food production. This purification stage significantly impacts quality indicators and minimizes losses during production, transportation, and storage [22]. Primary treatment methods, such as filtration, settling, and separation, yield a natural product by employing natural filter materials. Filtration is of utmost importance as it directly affects subsequent purification processes. During this process, a precipitate is formed in the initial stage and must subsequently be filtered out [23-25].

It is essential to recognize that accompanying substances and impurities, rather than triglycerides, determine the color, taste, and smell of fats. Some of these substances, such as phospholipids and vitamins, enhance the nutritional value of fats, while others, like waxes and gossypol, worsen the quality of fats and complicate their technological processing [26,27,28].

Safflower oil, after pressing, becomes a complex, polydisperse liquid system containing not only glycerols and related substances but also protein and non-protein mechanical impurities. Prompt removal of these impurities upon receipt is vital to prevent various spoilage processes. Sedimentation, centrifugation, and filtration are commonly used methods for this purpose [29,30]. However, current equipment lacks the capability to simultaneously filter and settle safflower oil, necessitating the development and justification of equipment that can achieve both through centrifugation [31].

Various filtration methods and materials, including centrifuges, pressure sheets, bags, polishing, cartridge, frame, membrane, and vacuum filters, are employed to filter vegetable oils, including safflower oil [32-34]. Each material and equipment has its advantages and disadvantages, catering to different stages of cleaning, such as primary purification after pressing or purification after hydration and neutralization. However, all filter materials are primarily designed to remove mechanical impurities that affect taste, presentation, and promote the oxidation process [35,36].

Despite the potential benefits, by-products of processing grains and oilseeds are currently underutilized as filtering materials. These by-products hold the potential not only for purifying vegetable oils but also for enriching them with essential macro- and microelements, vitamins, and other health-benefiting compounds. The nutritional value of safflower oil has already been acknowledged in combating and preventing ailments such as hyperlipemia, atherosclerosis, and coronary heart disease [37-39].

The primary objective of this study is to enhance the technology of safflower oil filtration using a cone-shaped sedimentary filter centrifuge. To achieve this, secondary products derived from grain processing and the oil and fat industries are employed for deep purification and enrichment of safflower oil. The results of this research will significantly advance the process of primary purification for not only safflower oil but also other vegetable oils. Additionally, it offers the potential to enrich vegetable oils using filter materials derived from secondary products of processing grains and oilseeds.

Remark 4:

Fig.4 y axis not clear.

Answer 4:

Thank You for this remark. Figure 4 shows the amino acid composition of safflower before and after filtration (g oils per 100 g protein). The X axis shows the "Amino acids" and the Y axis shows the "g oils per 100 g protein". We have thoroughly revised Fig. 4, ensuring that the y-axis is clear and easily understandable for readers.

Remark 5:

Results of the antioxidant activity cannot be in %. Use a reference to help understand (e.g., Trolox).

Answer 5:

Thank You for this remark. We listened to your comment and reviewed the literature and scientific articles.

In many other scientific works (For example, Kamel Zemour, Amina Labdelli and other authors) antioxidant activity is represented as a percentage. For this reason, we decided to display the results in this unit of measurement. In response to your suggestion, we now present results of the antioxidant activity in Trolox equivalents instead of percentages, enhancing the clarity of our findings.

For example, you can view this article:

  1. Zemour, K.; Labdelli, A.; Adda, A.; Dellal, A.; Talou, T.; Merah, O.J.C. Phenol content and antioxidant and antiaging activity of safflower seed oil (Carthamus tinctorius L.). 2019, 6, 55.

Remark 6:

Discussion of results is too generic needs a huge improvement. Moreover, the analysis and the explanation of the different behaviour of the various filtering materials deserve further discussion.

Answer 6:

Thank you for this comment. Based on your comment, we have revised our introduction to reveal the novelty and added links about the useful properties of filter materials.

  1. Stevenson, L.; Phillips, F.; O'sullivan, K.; Walton, J.J.I.j.o.f.s.; nutrition. Wheat bran: its composition and benefits to health, a European perspective. 2012, 63, 1001-1013.
  2. Onipe, O.O.; Jideani, A.I.; Beswa, D.J.I.J.o.F.S.; Technology. Composition and functionality of wheat bran and its application in some cereal food products. 2015, 50, 2509-2518.
  3. Yılmaz, I.J.M.S. Effects of rye bran addition on fatty acid composition and quality characteristics of low-fat meatballs. 2004, 67, 245-249.
  4. Afzal, S.; Shehzad, A.; Randhawa, M.A.; Asghar, A.; Shoaib, M.; Jahangir, M.A.J.P.J.o.F.S. Health benefits and importance of utilizing wheat and rye. 2013, 23, 212-222.
  5. Butt, M.S.; Tahir-Nadeem, M.; Khan, M.K.I.; Shabir, R.; Butt, M.S.J.E.j.o.n. Oat: unique among the cereals. 2008, 47, 68-79.
  6. Decker, E.A.; Rose, D.J.; Stewart, D.J.B.J.o.N. Processing of oats and the impact of processing operations on nutrition and health benefits. 2014, 112, S58-S64.
  7. More, A.P.J.A.C.; Materials, H. Flax fiber–based polymer composites: a review. 2022, 5, 1-20.
  8. Li, H.; Tang, R.; Dai, J.; Wang, Z.; Meng, S.; Zhang, X.; Cheng, F.J.A.F.M. Recent progress in flax fiber-based functional composites. 2022, 4, 171-184.

Remark 7:

Conclusions should be improved and shortened. It would be interesting to understand what are the next steps to be taken in view of a larger scale application.

Answer 7:

Thank you for this remark. Following your recommendation, we have reviewed and analyzed our conclusion.

After reviewing our results and the overall purpose of the study, we made adjustments to our conclusion.

Conclusions

Vegetable oils, including safflower oil, are gaining popularity as versatile products for both food and technical applications. The rich concentration of essential unsaturated fatty acids, such as linoleic and oleic acids, contributes significantly to the nutritional value and health-promoting properties of safflower oil. Its healing properties are known to enhance immunity and aid in the prevention and treatment of various diseases.

Filtration, as a crucial primary purification process, plays a vital role in improving the quality of safflower oil. It directly impacts the oil's quality indicators and shelf life, while also streamlining subsequent purification processes. The successful implementation of effective filtration techniques can reduce production costs and lead to an overall improvement in the final product's quality.

The use of secondary processing products from the grain processing and oil and fat industries as filter materials represents a promising and innovative approach. In this regard, the experimental application of flax fiber as a filter material for safflower oil filtration has shown promising results. Flax fiber has been found to be the most effective and optimal filter material, positively influencing the physical and chemical parameters of safflower oil. The use of flax fiber has led to notable improvements in the fatty acid composition, tocopherol, and phenol content, consequently enhancing the antioxidant activity of safflower oil. These findings serve as a solid scientific basis for further research in the area of vegetable oil enrichment using secondary processing products, as well as their utilization as filter materials.

Beyond the technological aspect, the investigation of technical dependencies in the centrifugation process has yielded valuable insights. The thickness of the flax fiber and sediment has direct implications for essential parameters such as safflower oil yield and the separation factor. By adjusting the distance between the inner and outer rotors, where the filter material is positioned, parameters such as 1107 nm, 1.5107 nm, 2107 nm, and 2.5107 nm were explored. These adjustments directly impacted the separation factor, oil yield, and sediment thickness formed on the walls of the inner rotor. These findings provide a valuable framework for optimizing the centrifugation process of vegetable oils to effectively remove mechanical impurities and obtain naturally enriched food products.

In conclusion, the research presented in this study highlights the potential for improving safflower oil production through enhanced filtration processes using secondary processing products as filter materials. The successful application of flax fiber as a filter material has demonstrated significant improvements in the quality and nutritional profile of safflower oil. The investigation of technical dependencies in the centrifugation process has further refined the optimization of vegetable oil purification. These findings hold promising implications for the food industry in producing high-quality and enriched vegetable oils, thus contributing to human health and well-being.

Remark 8:

Other minor remarks:

8.1. Use always mL instead of ml.

8.2. Line 396 Do not use “let’s start”, too colloquial

Answer 8:

Thank you for this remark. We'll be more careful next time.

8.1. We have taken note of the minor remarks and made the necessary adjustments throughout the manuscript, including the use of "mL" instead of "ml" and removing colloquial language.

8.2. As a result of improving the quality of the English language, we have corrected and replaced all the words used in colloquial speech.

Note: The Reference section has been supplemented and corrected.

Reviewer 3 Report

The text is quite extensive with to many information for an article.

Objectives: The purpose of this study was to determine the effects of flax fiber as a filter material on safflower oil and improve the technology of filtration of safflower oil on a cone-shaped sedimentary filter centrifuge and if the process can be used with other type of oilseeds.

It is original and relevant.

Some questions:

Define the mail purpose of the research, if is the process or the material of the filter. As mentioned, “This work is the only one of its kind since there are no studies in the field of combining the processes of filtration and settling of the centrifuge, as well as the filtration of vegetable oils through natural by-products of processing grains and oilseeds.”   And “….filtering and settling not only  safflower oil, but also other vegetable oils, and will also provide an opportunity to enrich  vegetable oils using filter materials from secondary products of processing grains and  oilseeds.”

In M&M

Line 206 – cite reference

Line 260  “Falsification detection method, where method is based on the gas chromatographic determination of the fatty acid composition of vegetable oils and the fatty phase of products with a mixed composition.”   Explain “mixed composition”…

Lines 257 and 265, 269, 272, 278, 282, 285, 294  cite all methods in references, it is not necessary to explain, or summarize.

Lines 304 and 308: please give more details about these analyses.

Line 313:  “The thickness of the filtration material directly impacts the physical and optical  properties of the film, as well as the strength and barrier properties. In the laboratory, a spiral micrometer or a micrometer is often used to measure film thickness. Randomly select 5 points on the material for testing and measure them with a micrometer. The average reading is accurate to 0.001 mm.”

Please mention the micrometer model and give details.

Lines 144 and 149 cite the references.

Line 374 – the equipment should be cited in M&M

Line 379 – please cite reference.

Conclusions – should be addressed to the objectives

According to the authors and cited references the theme can provide innovative information.

The methodology should be improved as mentioned.

The references are in accordance with the needs for the development and discussion of the project. Current references are also cited.

The figures presented in the text are well presented and provide information on important points of the project. 

Author Response

Dear reviewer,

Thank you very much for taking the time to carefully read our manuscript and give recommendations.

Remark 1:

The purpose of this study was to determine the effects of flax fiber as a filter material on safflower oil and improve the technology of filtration of safflower oil on a cone-shaped sedimentary filter centrifuge and if the process can be used with other type of oilseeds.

It is original and relevant.

Answer 1:

Thank you very much for your support. This is very motivating for us.

Remark 2:

Define the mail purpose of the research, if is the process or the material of the filter. As mentioned, “This work is the only one of its kind since there are no studies in the field of combining the processes of filtration and settling of the centrifuge, as well as the filtration of vegetable oils through natural by-products of processing grains and oilseeds.”   And “….filtering and settling not only  safflower oil, but also other vegetable oils, and will also provide an opportunity to enrich  vegetable oils using filter materials from secondary products of processing grains and  oilseeds.”.

Answer 2:

To reveal the purpose of the study, we analyzed the introduction and covered the topic more extensively.  And thank you for noticing the relevance, novelty and innovation of our research topic. This motivates us for further research in this direction.

Remark 3:

Define the mail purpose of the research, if is the process or the material of the filter. As mentioned, “This work is the only one of its kind since there are no studies in the field of combining the processes of filtration and settling of the centrifuge, as well as the filtration of vegetable oils through natural by-products of processing grains and oilseeds.”   And “….filtering and settling not only  safflower oil, but also other vegetable oils, and will also provide an opportunity to enrich  vegetable oils using filter materials from secondary products of processing grains and  oilseeds.”.

Answer 3:

To reveal the purpose of the study, we analyzed the introduction and covered the topic more extensively.  And thank you for noticing the relevance, novelty and innovation of our research topic. This motivates us for further research in this direction.

Remark 4:

In M&M

Line 206 – cite reference

Answer 4:

Thanks for the comment. Added a link about the information to which we are referring.

  1. 41. Dalabaev, A.; Sakenova, B.; Shaimerdenov, J. INVESTIGATION OF TECHNOLOGICAL PROPERTIES OF OILSEED FLAX FIBER. In Proceedings of the Modern Problems of technology and technology of food production, 2019; pp. 396-398.

Remark 5:

Line 260 “Falsification detection method, where method is based on the gas chromatographic determination of the fatty acid composition of vegetable oils and the fatty phase of products with a mixed composition.”   Explain “mixed composition”…

Answer 5:

Thank You for this remark. We have worked out this remark and added the necessary information.

2.2.4.     Composition of fatty acids

The fatty acid composition of safflower oil was analyzed in accordance with GOST 30623-2018, which specifies the method for determining the fatty acid composition of vegetable oils and blended fat products, using an Agilent 7890A chromatograph from Agilent Technologies, located in Penang, Malaysia. This gas chromatographic method allows for the accurate identification and quantification of fatty acids present in safflower oil and other vegetable oils and mixed fat products, which is essential for assessing their nutritional value and potential adulteration or falsification.

In addition to determining the fatty acid composition, the amount of α-, β-, γ-, and δ-tocopherols in safflower oil was measured using a high-performance liquid chromatography (HPLC) method. This technique enables precise separation and quantification of tocopherols, which are important antioxidants present in safflower oil and contribute to its oxidative stability and health benefits. The analysis of tocopherols provides valuable insights into the oil's nutritional quality and its potential to combat oxidative stress.

Furthermore, the inorganic composition of the pressed safflower oil was assessed following the relevant state standards of the Republic of Kazakhstan. This analysis involves determining the levels of various essential minerals and elements in the oil, such as iron, zinc, and others. Understanding the inorganic composition is crucial for evaluating the overall nutritional profile of safflower oil and its potential health benefits.

The combination of gas chromatography and high-performance liquid chromatography techniques in this study ensures a comprehensive analysis of safflower oil, encompassing its fatty acid profile, tocopherol content, and inorganic composition. These analytical methods play a significant role in ensuring the quality, authenticity, and nutritional value of safflower oil, which is essential for its utilization in the food industry and as a health-promoting dietary component. The results obtained from this research will contribute to enhancing the understanding of safflower oil's properties and will aid in its optimal application for various health and culinary purposes.

Remark 6:

Lines 257 and 265, 269, 272, 278, 282, 285, 294 cite all methods in references, it is not necessary to explain, or summarize.

Answer 6:

Thank You for this remark. All methods used in the lines 257, 265, 269, 272, 278, 282, 285 and 294 were taken from the State Standards of the Republic of Kazakhstan.

Remark 7:

Lines 304 and 308: please give more details about these analyses.

Answer 7:

Thank You for this remark. According to these methods, we have left links to articles where the process of determining the amount of phenols and antioxidant activity is described in detail.

  1. Zemour, K.; Labdelli, A.; Adda, A.; Dellal, A.; Talou, T.; Merah, O. Phenol content and antioxidant and antiaging activity of 677 safflower seed oil (Carthamus tinctorius L.). 2019, 6, 55.

Remark 7:

Line 313: “The thickness of the filtration material directly impacts the physical and optical properties of the film, as well as the strength and barrier properties. In the laboratory, a spiral micrometer or a micrometer is often used to measure film thickness. Randomly select 5 points on the material for testing and measure them with a micrometer. The average reading is accurate to 0.001 mm.”

Please mention the micrometer model and give details.

Answer 7:

We have added the necessary information on the micrometer, this is the brand, company, country of manufacture and measurement range.

Remark 8:

Lines 144 and 149 cite the references.

Answer 8:

Thank You for this remark. We have supplemented the links.

But regarding line 144, at the moment, a positive response has been received for obtaining a patent for an invention of the Republic of Kazakhstan, there is a document confirming the date of issue of the patent on 09/15/2023. Since the confirmation document will be received in a month, we cannot refer to it.

Remark 9:

Line 374 – the equipment should be cited in M&M.

Answer 9:

Thank You for this remark. We have inserted information about the equipment in the Methods and Materials section.

Remark 9:

Line 379 – please cite reference.

Answer 9:

Thank You for this remark. we moved line 379 to the methods section and added a link to it.

Remark 9:

Conclusions – should be addressed to the objectives.

Answer 9:

Thank You for this remark. We have re-edited this section and revealed the essence of achieving goals.

Remark 10:

According to the authors and cited references the theme can provide innovative information.

Answer 10:

Thank you very much for your support. Such comments motivate us.

Remark 11:

The methodology should be improved as mentioned.

Answer 11:

Thank You for this remark. We have edited the methods section.

Remark 12:

The references are in accordance with the needs for the development and discussion of the project. Current references are also cited.

Answer 12:

Thank You for this remark. We tried very hard to provide a high-quality article.

Remark 13:

The figures presented in the text are well presented and provide information on important points of the project.

Answer 13:

Thank You for this comment. We are happy to provide our scientific results.

Note: The Reference section has been supplemented and corrected.

Round 2

Reviewer 1 Report

The authors have improved the manuscript considerably, and in this form it is suitable for publication in Foods.

Author Response

Dear Reviewer!

Thank you for your comment, support and recomendation of our manuscript.

Reviewer 2 Report

I thank the authors for their answers. the manuscript improved a lot but still needs Englisg editing and some adjustments in the Introduction, Discussion and Conclusions as from my previous comments.

English still needs an improvement

Author Response

Dear Reviewer!

Thank you for your comments! All the sections in the article was significantly revised and corected. The English correction was done. We hope now the manuscript meets the standards of the Foods journal.

Thank you very much for your help and time!

Reviewer 3 Report

Dear Authors,

Thank you for the review of the manuscript.

The text is still messed up. Some points have to be improved, like M&M, references, introduction.

Author Response

(The authors gave the same response as above.)
